# Neutrophil to Lymphocyte Ratio as a Biomarker for the Prediction of Cancer Outcomes and Immune-Related Adverse Events in a CTLA-4-Treated Population

**DOI:** 10.3390/cancers17122011

**Published:** 2025-06-17

**Authors:** Michael M. Cunningham, Rachel Romero, Carolina Alvarez, Shruti Saxena Beem, Todd A. Schwartz, Rumey C. Ishizawar

**Affiliations:** 1Division of Rheumatology, Allergy, and Immunology, Department of Medicine, University of North Carolina, Chapel Hill, NC 27514, USA; rumey_ishizawar@med.unc.edu; 2Atrium Health Rheumatology, 200 Medical Park Drive, Suite 330, Concord, NC 28025, USA; rachel.romero@atriumhealth.org; 3Thurston Arthritis Research Center, University of North Carolina, Chapel Hill, NC 27599, USA; alvarec@live.unc.edu (C.A.); tschwart@email.unc.edu (T.A.S.); 4Department of Biostatistics, Gillings School of Global Public Health, University of North Carolina, Chapel Hill, NC 27599, USA

**Keywords:** immune-related adverse events, immune checkpoint inhibitor, immunotherapy, cancer outcomes, cancer biomarker, rheumatologic irAEs, CTLA-4 blockade, neutrophil to lymphocyte ratio, predictive modeling

## Abstract

This study looked at whether side effects of a type of cancer therapy called immunotherapy could be predicted by a blood test and whether these side effects could help predict who would better respond to the treatment. The test is called a “neutrophil-to-lymphocyte” ratio (NLR) and can be obtained from a simple blood draw. Researchers reviewed data from 111 patients who had cancer, mostly a type of skin cancer called melanoma, and were treated with a specific immunotherapy, ipilimumab, to answer these questions. Patients who had multiple side effects tended to have a better cancer outcome. Additionally, patients with low NLR levels were more likely to experience side effects. This suggests that the NLR could be a useful and low-cost tool for predicting treatment responses and side effects in patients receiving this cancer therapy. More research is needed to confirm these results in larger groups and with other types of cancer.

## 1. Introduction

Immune checkpoint inhibitor (ICI) therapeutics is a growing area of cancer immunotherapy that serves to induce anti-tumor immune responses via a blockade of immune checkpoint proteins, namely, programmed cell death protein 1 (PD-1) and cytotoxic T-lymphocyte associated protein 4 (CTLA-4) [1]. Additional ICI targets are under investigation with the more recent approval of anti-LAG3 [2,3,4]. The growing number of ICIs indicates the effectiveness of this class of cancer therapy. Immune checkpoints play a key role in the maintenance of self-tolerance [5]. Tumor cells utilize immune checkpoint pathways to evade anti-tumor targeting by immune system proteins [1,6]. To combat this, checkpoint inhibitors impede this pathway. While obstruction of this regulatory pathway has led to efficacy in cancer treatment, this disinhibition of the immune system is not tumor specific and can lead to autoimmune targeting of healthy host tissue, termed immune-related adverse events (irAEs) [1,7]. While generally felt to be better tolerated than older chemotherapeutic regimens [8], these irAEs can be associated with significant morbidity, irreversible organ damage, and in rare incidences mortality, of cancer patients [9,10].

Given the potential for significant toxicity, multiple biomarkers have been investigated to predict risk for irAEs [11,12,13]. One such biomarker is the peripheral blood neutrophil to lymphocyte ratio (NLR). A low NLR (i.e., a higher proportion of lymphocytes) is associated with cancer outcomes [14,15]. While the underlying mechanistic pathways are not well elucidated, there is some thought that the peripheral leukocyte differential could mirror tumor microenvironments and that significant neutrophilia can inhibit the cytolytic function of lymphocytes, thereby neutralizing anti-tumor effects [14]. As the therapeutic mechanisms of ICI therapy are mediated by lymphocytic activity, the NLR was investigated as a potential predictor for response to therapy. To date, few studies have shown that a low NLR has been associated with improved cancer outcomes in those treated with ICI therapy, largely with PD-1 [11,16,17,18,19,20,21,22,23,24]. Interestingly, irAEs are also associated with improved cancer outcomes in ICI-treated patients, suggesting a possible mechanistic link [1,5,16,17,18,19]. This led to the NLR being investigated as a predictor for irAEs, where recent data has shown a lower NLR to be associated with irAEs among PD-1-treated patients [17,18,19,25]. Less is known about the CTLA-4 blockade, which acts primarily in earlier stages of T cell activation when compared to PD-1, which acts after T cell activation [25,26,27,28]. Furthermore, CTLA-4 has a more defined effect on T-regulatory cells and thus may have greater potential to alter lymphocyte populations. Lastly, CTLA-4 and PD-1 are expressed in varying amounts, depending on the organ system. These differences may account for the distinct risk profiles of certain organ-specific irAEs (e.g., PD-1 has been associated with thyroid dysfunction, while the CTLA-4 blockade more commonly causes hypophysitis) [29,30].

This study investigates a cohort of patients treated with an anti-CTLA-4 ICI, ipilimumab, and characterizes the irAEs, evaluates irAE incidence as a predictor for improved cancer outcome, and examines whether a low NLR predicts irAE occurrence and cancer outcome.

## 2. Materials and Methods

IRB approval (IRB 17-1841) was obtained prior to any research activity. We conducted a single-institution retrospective electronic medical record review of patients being treated with ipilimumab by the University of North Carolina at Chapel Hill Division of Oncology from 1/2004 to 7/2017. Ipilimumab was chosen, as it was the only CTLA-4 ICI therapy offered at this site during the timeframe studied. All patient charts were independently reviewed by a single physician. In areas of uncertainty, a second physician reviewer was used to determine clinical outcomes.

The type of ICI therapy was defined as ipilimumab monotherapy or ipilimumab in combination with another ICI. irAE occurrence was based on the patient’s oncologist or other subspecialist assessment determined by a review of clinical documentation and defined as outlined in Appendix A; these irAEs were organized by organ involved (e.g., GI/HEP for gastroenterology or hepatic manifestations). While the cohort did have cardiac, pulmonary, and neurological events, these were rare with *n* < 5 and were thus excluded from the final analysis. When irAEs from multiple organ systems were identified, these were categorized as involving one or more, two or more, or three or more systems involved. Three groups of cancer response were defined: no evidence of disease/complete remission (NED/CR), minimal residual disease/stable disease (MR/SD), and progression of disease (PD). Cancer response was evaluated from the most recent known oncologic evaluation at the time of chart review. ECOG status and cancer stage were determined by the cancer care team as standard care and were abstracted without change from electronic medical records.

To determine the NLR, baseline complete blood counts (CBCs) were recorded prior to the initiation of ICIs for all patients. All patients had recorded CBCs; these were primarily on the day of treatment if not the day before. There is no accepted standard value for a “low” NLR. In this study, the values of <4 and <5 were each investigated based on previously published cutoffs in PD-1/PD-L1-treated patients, as there were very limited data for CTLA-4-treated patients [16,17,18,19]. The majority of CBC tests were performed at UNC’s McLendon Laboratories Clinical Lab (Chapel Hill, NC, USA), with the remaining values obtained from outside entities (e.g., commercial labs).

Descriptive statistics were used to summarize the study patients and relevant variables. Frequencies (percentages) were produced for categorical variables, while mean (standard deviation (SD) and range) were computed for continuous variables. Separate, multivariable logistic models were used to model the following outcomes: (1) irAE incidence by type and (2) whether a threshold for the number of irAEs was exceeded. Consideration of irAEs by type was limited to organ types with more than 5 events in the sample. Separate, multivariable, partial proportional odds models (with distinct parameter estimates for sex due to this covariable showing a significant (*p* < 0.05) effect) were used to model the outcome (3) with three-level ordinal cancer response among patients with cancer, with probabilities in the logistic model cumulated for better versus worse cancer response. Odds ratios (ORs) and 95% confidence intervals (CIs) were produced to characterize the association between the baseline NLR < 4 or NLR < 5 and each outcome stated above. Receiver operating characteristic (ROC) curve analysis was used to assess the appropriateness of these NLR literature thresholds to our sample. Hazard ratios (HRs) and 95% confidence intervals (CIs), using time-dependent Cox proportional hazard models, were produced to characterize the association between irAE definitions and either time to PD or time to mortality. Violations of the proportional hazards assumption by covariables other than the main effect (i.e., irAE definition) resulted in models stratified by that covariable. All models were adjusted for age, sex, and ICI therapy, and a second set of models was additionally adjusted for ECOG score and cancer stage at the start of ICI therapy. In an exploratory fashion, interactions between NLR definition with age, sex, and ICI therapy (effect modifiers) were tested separately for all models. Results are produced for each level of an effect modifier if it is found to be significant at the 0.10 level, which was chosen as a threshold to detect at least modest effect modification.

All analyses were performed with SAS version 9.4 (SAS institute Inc., Cary, NC, USA). Statistical significance was determined at an alpha level of 0.05, except where otherwise noted.

## 3. Results

We identified 116 patients who received ipilimumab during the defined time period. The mean age for the cohort was 57 years of age with a balanced sex distribution. Clinically, while half the cohort had a baseline ECOG status of 0 (n = 61; 55%), the concurrent cancer staging was either at stage 3 (n = 49; 44.1%) or at stage 4 (n = 59; 53.2%) (see Table 1).

The predominant cancer diagnosis was melanoma (n = 100; 91.3%), but the cohort also included lung and bladder cancer patients. Of the 116 patients initially identified, 5 were excluded as they had received triple ICI therapy (pembrolizumab, nivolumab, and ipilimumab). Of the 111 remaining, 36% received ipilimumab monotherapy, while the rest were in combination with another ICI. A total of 72% of these patients experienced at least one irAE of any type (see Table 2).

Based on previously published data showing an association between irAEs and improved cancer outcomes in the PD-1/PD-L1 blockade [1,5], we first looked at this association in our population of CTLA-4 ICI-treated patients. Among the 100 melanoma patients treated with CTLA-4 ICI therapy (ipilimumab), having more than one irAE was associated with better cancer outcomes (Figure 1). This inverse relationship of improved cancer outcomes is particularly demonstrated as a linear effect for an increase in one additional irAE (1.48 [1.02, 2.15], Table 3). This relationship remains when additionally adjusting for ECOG status and cancer stage at the time of ICI start (1.71 [1.13, 2.57], Table 3). Our data examining CTLA-4-directed ICI replicate findings where the development of the irAE is associated with improved cancer outcomes in PD-1 and PD-L1 ICI-treated patients [31].

There were varied associations found among organ-specific irAEs and improved cancer response, as shown in Table 3. Endocrine irAEs were statistically significantly associated with improved cancer outcomes (OR 2.82 [1.19, 6.67]) when adjusted for age, sex, and ICI therapy. The effect of improved cancer outcome was evidenced with endocrine irAEs with additional adjustment for ECOG score and cancer stage (OR 2.51 [0.95, 6.65] but did not achieve statistical significance, likely due to reduced sample size. While other organ systems assessed (GI, Derm, Rheum/MSK) did not demonstrate a statistically significant association, likely due to limited statistical power, Rheum/MSK irAEs notably have a suggestion of association with improved cancer outcomes (OR 2.56 [0.63, 10.3]) when adjusted for age, sex, and ICI therapy that is even further enhanced with adjustment for ECOG score and cancer stage with an OR 4.46 (0.90, 22.2). However, the adjusted odds ratio for Rheum/MSK irAEs with cancer response is imprecise, as seen by the wide confidence interval. This reflects the limited number of Rheum/MSK events (n = 8) (Table 2).

Evaluating time-to-event outcomes of mortality and disease progression in this cohort revealed that irAE occurrence was associated with a lower risk of either when adjusted for ECOG score and cancer stage (mortality HR 0.49 [0.26, 0.93] and disease progression HR 0.40 [0.20, 0.82], Table 4). We then stratified for one irAE or more than one irAE, i.e., two to four organ systems involved, as no patient had more than four organ systems involved. Having more than one irAE was associated with a more beneficial outcome (HR 0.52 [0.27, 0.99], Table 4). Evaluating organ-specific irAEs suggested a possible association with a lower rate of PD and mortality, but none were found to be statistically significant.

When including the cancer stage at the start of the ICI, this variable violated proportional hazards so that the results were stratified by cancer stage. These results showed that the rate of disease progression after having a first irAE (vs. no irAE) significantly decreased, particularly among patients who had stage 4 disease (HR 0.36 [0.14, 0.89], Appendix A). The association was also strongest in those with 2–4 irAEs (vs. none) (HR 0.24 [0.08, 0.75], Appendix A).

To better predict patients at risk for irAEs and cancer outcomes, the NLR was investigated for associations with irAE incidence. A low NLR has been found to be associated with irAE incidence and improved cancer outcomes in PD-1/PD-L1-treated patients, but less has been published on CTLA-4-treated patients. We found that a low NLR was associated with increased odds of irAEs of any organ type or severity in patients treated with ipilimumab (Table 4). The effect was larger using a cutoff value of NLR < 5 (OR 4.34 [1.73, 10.9]) as opposed to NLR < 4 (OR 2.35 [0.99, 5.58]). These effects also appear within the range of previously published data in PD-1/PD1-L1 ICI-treated patients who largely had non-small cell lung cancer (NSCLC) [16,17,18,19].

When stratifying irAEs by organ system, there was no statistically significant association with a low NLR found (Table 5). Despite this, these data show a potentially noteworthy effect for NLR < 5 with Rheum/MSK irAEs (4.40 [0.53, 36.9]). In addition, we noted a differing association of a low NLR with dermatologic irAEs with regard to sex, with a suggestion (*p* < 0.1) of an association of a low NLR with DERM among women but not men (for NLR < 4: 4.41 [1.17, 16.6] and NLR < 5: 12.2 [1.40, 106] among women and 0.90 [0.32, 2.49] and 1.04 [0.34, 3.14] among men, respectively), as shown in Appendix A. While exploratory, this heterogeneity between sexes is an intriguing finding not previously reported.

Lastly, a low NLR was examined for association with improved cancer outcomes. This hypothesis stems from the data above, demonstrating that a low NLR is associated with irAEs, and irAEs have been associated with improved cancer outcomes. In this population, there was a trend that a low NLR is associated with improved cancer outcomes but did not meet statistical significance (Appendix A; OR 0.76 [0.36, 1.59] for NLR < 4).

We originally used NLR cutoffs of 4 and 5 based on previous data on PD-1 inhibition, with less guidance in a CTLA-4 cohort due to limited published data on CTLA-4 as related to the NLR and irAEs. When considering receiver operator characteristic curve optimization to identify the ideal NLR cutoff to predict any irAE occurrence (Figure 2), NLR < 5.09 was determined to be the optimal threshold value and indicates that the literature cutoffs considered are appropriate in our cohort, particularly NLR < 5.

## 4. Discussion

In this population of anti-CTLA-4-treated patients, an association between additional irAEs and improved cancer outcomes was observed. The previous literature has shown similar findings among PD-1-treated patients [16,17,18,19,32], consistent with the well-described hypothesis of a mechanistic link between irAEs and anti-tumor effects [1]. To further characterize this association, categorical groups of irAE number were analyzed against cancer outcome. There was a suggestion for an association between an increased number of irAEs and improved cancer outcomes (Figure 1). Further investigation into time-to-event analysis (Table 4) revealed that irAE occurrence was linked to a lower risk of both mortality and disease progression. In subgroup analysis, this was found to be strongest among patients with stage 4 disease at the time of ICI initiation (Appendix A). There have been suggestions in the literature of certain organ-specific irAEs portending a better prognosis, most consistently with dermatologic irAEs and improved cancer outcomes in PD-1 ICI-treated patients [1]. Less is known for the CTLA-4 blockade.

When analyzing this cohort for organ-specific irAEs and cancer outcomes, only the association with endocrine irAEs was statistically significant. Endocrine irAEs have been associated with improved cancer outcomes in previously published works. As each organ-specific irAE may have different pathophysiological mechanisms, it has been suggested that endocrine irAEs may share a mechanism with the anti-tumor effect of ICI therapy [33]. In support of this previous finding, our data also supports the association of endocrine irAEs with improved cancer outcomes (OR 2.82 [1.19, 6.67]). Interestingly, Rheum/MSK, when compared to derm or GI, had a high OR but did not meet statistical significance, likely due to a low number of events. Of the many biomarkers that have been investigated to predict irAEs, the NLR has shown promise [11,20], although primarily for those treated with PD-1/PD-L1. Due to limited publications regarding NLR in CTLA-4-treated cancer patients, we investigated whether a low NLR value was associated with an increased incidence of irAEs in this CTLA-4-treated cohort. Our data revealed that in a CTLA-4-treated cohort, a low NLR is associated with higher odds of developing an irAE. The effect sizes that we saw in this cohort are similar to what has been seen in the PD-1 blockade [16,17,18,19]. There have been varying degrees of association reported between the pre-treatment NLR and organ-specific irAEs, leading to hypotheses about different pathogenic mechanisms playing key roles in each irAE [1,20]. When the pre-treatment NLR was analyzed against organ-specific irAEs in this cohort, the data suggested a few trends. There was a suggestion of a low NLR being associated with Rheum/MSK irAEs (Table 4). This is hypothesis generating, and a larger cohort of these patients will be needed to further clarify this relationship. When analyzing the data for sex differences, in an exploratory fashion, there was a difference among dermatologic irAEs, with a low pre-treatment NLR appearing to be more strongly associated with increased risk amongst female participants (Appendix A). Data on sex differences in irAE incidence and organ manifestations has been mixed in the literature and not well established; however, multiple studies have suggested a potential difference [31].

Our retrospective cohort study characterizing irAEs amongst patients treated with anti-CTLA-4 ICI therapy and investigating the NLR as a potential biomarker for prediction showed similar findings to that in PD-1/PD-L1 ICI-treated patients. A lower NLR was shown to be associated with irAEs, and additional irAEs were found to be associated with improved cancer outcomes. Moreover, our study supports using a cutoff for NLR < 5 in receiver operating characteristic curve analysis (Figure 2). This further adds to the growing body of literature that shows that a lower NLR could be a peripheral biomarker for the prediction of irAEs and cancer outcomes amongst those treated with either class of ICI (PD-1/PD-L1 or CTLA-4). Using the NLR would provide oncologists with additional data to help predict irAEs and guide monitoring strategies for patients. For example, if a patient has a lower NLR and, therefore, a higher risk for an irAE, they may increase monitoring frequency or be quicker to escalate investigation into symptoms that may be harbingers of an impending irAE. As the NLR is calculated from a CBC with differential, it is readily available in most, if not all, clinical settings. However, further research with larger cohorts of varied cancer types is likely needed before such a biomarker could be implemented into clinical practice. Some limitations of the study include its limited scope, as most patients were treated with melanoma, and all CTLA-4-treated patients were treated with ipilimumab, raising questions about whether this truly is a class effect. As the study population was from 2004 to 2017 and the standard of care in oncology is rapidly changing, there is potential for temporal bias. Beyond larger validation studies, future studies of interest would include evaluating the NLR over time to see if there is a temporal change preceding an irAE. Such a finding would give providers more real-time information to predict irAEs. It would also be of interest to investigate NLR values and the severity of irAEs. Unfortunately, our cohort’s irAE grades were not readily available or consistently measured to investigate this question. We would be interested in investigating this in future studies.

## 5. Conclusions

This retrospective cohort study on CTLA-4-treated patients showed that immune-related adverse event incidence correlated with improved cancer outcomes, similar to that seen in PD-1-treated patients. A low NLR was associated with increased irAE incidence, which in turn was associated with improved cancer outcomes. However, a low NLR was not found to be associated with improved cancer outcomes. We hypothesize that this is from a limitation of power and, therefore, hypothesis generating. Further hypothesis-generating findings include a suggestion of NLR’s association with organ-specific irAEs, such as Rheum/MSK irAEs. The exploratory findings suggested sex differences in dermatologic irAEs. These results imply NLR’s potential as a low-cost predictive biomarker for irAEs in CTLA-4-treated patients. However, these results are not powered to fully answer these hypotheses, and larger studies are needed for validation.

## Figures and Tables

**Figure 1 cancers-17-02011-f001:**
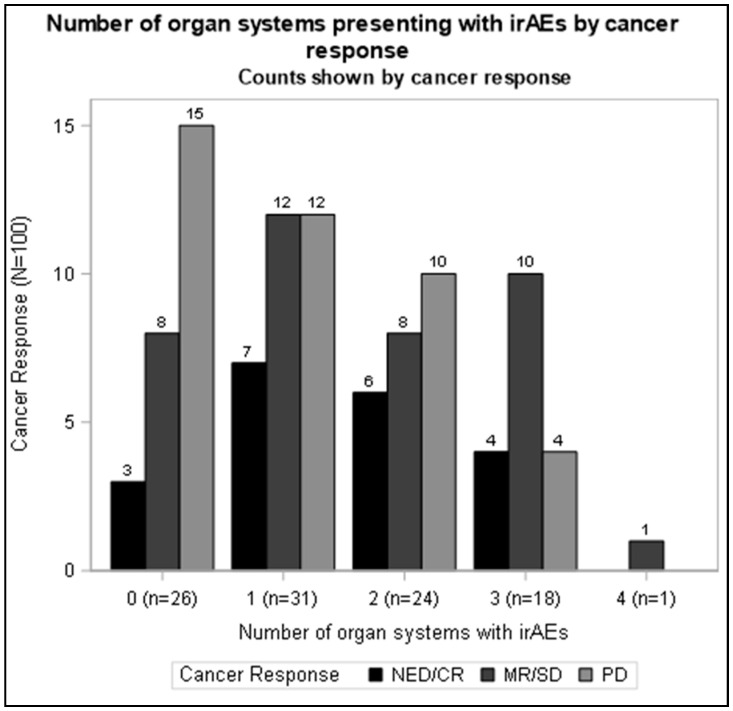
Number of organ systems affected by irAEs by cancer response. Cancer response is designated as no evidence of disease and/or complete response (NED/CR), marginal response and/or stable disease (MR/SD), or progression of disease (PD).

**Figure 2 cancers-17-02011-f002:**
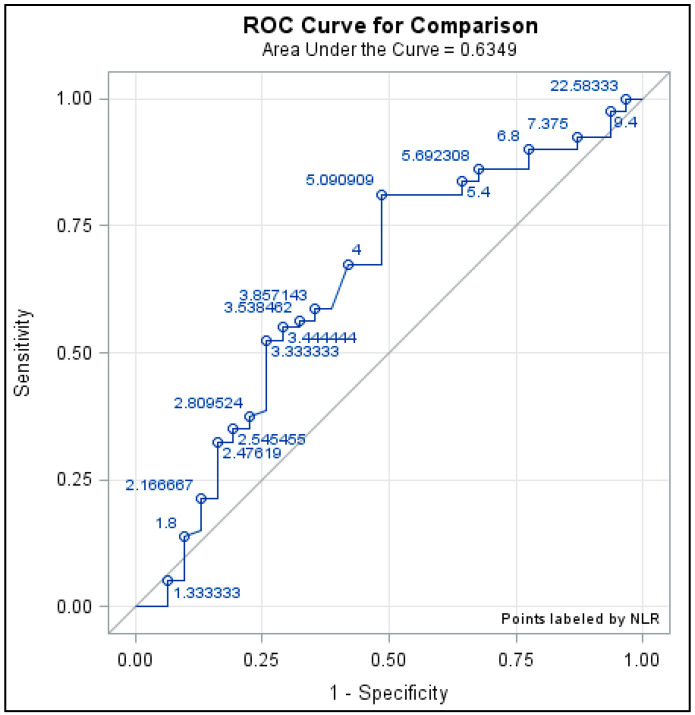
Receiver operator characteristic (ROC) curve for a comparison of NLR values to best predict irAEs.

**Table 1 cancers-17-02011-t001:** Descriptive statistics of baseline characteristics for patients with non-varying ICI therapy (N = 111) and a subgroup of patients only being treated for melanoma with an available cancer response (N = 100).

		Patients with Non-Varying ICI Therapy (N = 111)		Patients Treated for Melanoma with Cancer Response (N = 100)	
Baseline Characteristics		n *	% *	n *	% *
Age, mean (±SD) years (range = 17 to 81)		57.1	(±13.0)	56.7	(±13.2)
Female		49	44.1	42	42.0
NLR, mean (±SD) (range = 0.20 to 45.3)		4.62	(±4.95)	4.25	(±3.20)
NLR < 4 (lit)		59	53.2	55	55.0
NLR < 5 (lit)		78	70.3	71	71.0
ECOG status at the start of the ICI	(Missing, N/A)	6	5.4	6	6.0
	0	61	55	58	58.0
	1	36	32.4	28	28.0
	2	7	6.3	7	7.0
	3	1	0.9	1	1.0
Cancer stage at the start of the ICI	(Missing, limited/no formal staging)	2	1.8	-	-
	<3	1	0.9	1	1.0
	3	49	44.1	45	45.0
	4	59	53.2	54	54.0

* unless otherwise specified; ICI = immune checkpoint inhibitor; NLR = neutrophil to lymphocyte ratio; ECOG = Eastern Cooperative Oncology Group (ECOG) performance status; ±SD = standard deviation.

**Table 2 cancers-17-02011-t002:** Descriptive statistics of ICI treatment and clinical outcomes for patients with non-varying ICI therapy (N = 111) and a subgroup of patients only being treated for melanoma with an available cancer response (N = 100).

		Patients with Non-Varying ICI Therapy (N = 111)		Patients Treated for Melanoma with Cancer Response (N = 100)	
Characteristics		*n* *	% *	*n* *	% *
IPI only		40	36.0	38	38.0
IPI with NIVO		57	51.4	48	48.0
IPI with PEMBRO		14	12.6	14	14.0
irAE by organ type	GI/HEP	49	44.1	46	46.0
	DERM	43	38.7	42	42.0
	ENDO	32	28.8	29	29.0
	RHEUM/MSK	10	9.0	8	8.0
	PULM	4	3.6	3	3.0
	OPTHL	3	2.7	3	3.0
	NEURO	4	3.6	4	4.0
	RENAL	3	2.7	2	2.0
Count of organ systems with irAEs	0	31	27.9	26	26.0
	1	34	30.6	31	31.0
	2	25	22.5	24	24.0
	3	20	18.0	18	18.0
	4	1	0.9	1	1.0
At least 1 type of irAE		80	72.1	74	74.0
At least 2 types of irAEs		46	41.4	43	43.0
At least 3 types of irAEs		21	18.9	19	19.0
Cancer Response	(Missing, N/A)	1	0.9		
	NED/CR	20	18.0	20	20.0
	MR/SD	40	36.0	39	39.0
	PD	50	45.0	41	41.0
PD, mean (SE), days		740	(79)	802	(84)
Mortality		53	47.7	43	43.0
Mortality, mean (SE), days		1066	(63)	1142	(64)

* unless otherwise specified; IPI = ipilimumab; NIVO = nivolumab; PEMBRO = pembrolizumab; irAE = immune-related adverse event; GI/HEP = gastrointestinal and/or hepatic irAE; DERM = dermatologic irAE; ENDO = endocrine irAE; RHEUM/MSK = rheumatologic and/or musculoskeletal irAE; PULM = pulmonary irAE; OPTHL = ophthalmology irAE; NEURO = neurological irAE; RENAL = renal irAE; NED/CR = no evidence of disease or complete remission; MR/SD = minimal residual disease or stable disease; PD = progression of disease; SE = standard error.

**Table 3 cancers-17-02011-t003:** Association of irAEs (as counts or by organ type) with better cancer response using adjusted odds ratios and 95% confidence intervals, OR (95% CI), and N = 100 melanoma patients.

* *	OR (95% CI) ^1^	OR (95% CI) ^2^
Model by irAE count definition		
1 irAE increase (linear, one additional irAE)	**1.48 (1.02, 2.15)**	**1.71 (1.13, 2.57)**
irAE (categorical)		
1 (vs. none)	1.76 (0.60, 5.13)	1.90 (0.55, 6.50)
2 (vs. none)	2.86 (0.90, 9.07)	**3.83 (1.11, 13.3)**
3 or 4 (vs. none)	3.10 (0.94, 10.2)	**4.62 (1.24, 17.2)**
At least 1 (vs. none)	2.38 (0.96, 5.93)	**3.10 (1.11, 8.63)**
At least 2 (vs. 1 or none)	2.22 (0.98, 5.05)	**3.03 (1.24, 7.43)**
3 or 4 (vs. 2 or fewer)	1.83 (0.69, 4.87)	2.32 (0.82, 6.59)
Model by irAE organ type definition		
GI/HEP	1.39 (0.65, 2.99)	1.55 (0.62, 3.88)
DERM	1.19 (0.55, 2.57)	1.16 (0.50, 2.70)
ENDO	**2.82 (1.19, 6.67)**	2.51 (0.95, 6.65)
RHEUM/MSK	2.56 (0.63, 10.3)	4.46 (0.90, 22.2)

^1^ Cumulative logistic regression using partial proportional odds models with probabilities cumulated over the better cancer response (NED/CR vs. MR/SD vs. PD), adjusting for age, sex (allowing for unequal slopes for sex), and ICI therapy; ^2^ additionally adjusted for ECOG score (≥1 vs. 0) and cancer stage (4 vs. <4), sample size n = 94; significant results at alpha = 0.05 are shown in **bold**.

**Table 4 cancers-17-02011-t004:** Association of irAEs with time-to-event outcomes of mortality or disease progression using adjusted hazard ratios and 95% confidence intervals, HR (95% CI), and N = 111 patients.

	Mortality (n = 111)	Disease Progression (n = 109)	Mortality (n = 103)	Disease Progression (n = 101)
	HR (95% CI) ^1^	HR (95% CI) ^1^	HR (95% CI) ^2^	HR (95% CI) ^2^
irAE definition				
Any/first	0.58 (0.32, 1.03)	**0.55 (0.31, 0.97)**	**0.49 (0.26, 0.93)**	**0.40 (0.20, 0.82)**
irAE (categorical)				
1 (vs. none)	0.68 (0.33, 1.37)	0.81 (0.40, 1.64)	0.61 (0.28, 1.32)	0.62 (0.27, 1.43)
2–4 (vs. none)	**0.52 (0.27, 0.99)**	**0.42 (0.22, 0.83)**	**0.44 (0.22, 0.88)**	**0.31 (0.14, 0.68)**
Site-specific irAEs				
GI/HEP	0.87 (0.49, 1.57)	0.84 (0.46, 1.55)	0.77 (0.41, 1.45)	0.70 (0.35, 1.41)
DERM	0.65 (0.36, 1.17)	0.67 (0.36, 1.23)	0.72 (0.39, 1.30)	0.66 (0.35, 1.26)
ENDO	0.74 (0.37, 1.46)	0.52 (0.25, 1.11)	0.71 (0.35, 1.44)	0.44 (0.19, 1.01)
RHEUM/MSK	0.91 (0.32, 2.59)	0.60 (0.21, 1.73)	0.74 (0.22, 2.48)	0.52 (0.15, 1.74)

^1^ Cox proportional hazards regression modeling time to outcome, adjusting for age, sex, and ICI therapy; ^2^ additionally adjusted for ECOG score (≥1 vs. 0) and cancer stage (4 vs. <4); irAEs are included as time-varying covariables; significant results at alpha = 0.05 are shown in **bold**.

**Table 5 cancers-17-02011-t005:** Association of the neutrophil (N) to lymphocyte (L) ratio, the NLR, with an incidence of any irAE or by organ system using adjusted odds ratios and 95% confidence intervals, OR (95% CI), and N = 111 all patients.

		NLR < 4	NLR < 5	NLR < 4	NLR < 5
irAE Outcome	irAE Count	OR (95% CI) ^1^	OR (95% CI) ^1^	OR (95% CI) ^2^	OR (95% CI) ^2^
Any	80	2.35 (0.99, 5.58)	**4.34 (1.73, 10.9)**	2.54 (0.99, 6.54)	**4.11 (1.51, 11.2)**
GI/HEP	49	1.03 (0.48, 2.22)	2.03 (0.85, 4.83)	0.89 (0.39, 2.04)	1.61 (0.64, 4.05)
DERM	43	1.70 (0.77, 3.72)	2.22 (0.90, 5.48)	1.57 (0.68, 3.59)	2.24 (0.87, 5.79)
ENDO	32	1.28 (0.55, 3.00)	0.80 (0.32, 1.97)	1.46 (0.59, 3.63)	0.82 (0.31, 2.15)
RHEUM/MSK	10	1.35 (0.35, 5.16)	4.40 (0.53, 36.9)	1.56 (0.34, 7.25)	3.52 (0.40, 31.3)
2 or more organ systems	46	1.21 (0.54, 2.68)	1.66 (0.69, 4.01)	1.08 (0.47, 2.51)	1.43 (0.57, 3.61)
3 or 4 organ systems	21	1.12 (0.42, 3.02)	1.86 (0.59, 5.87)	1.00 (0.34, 2.97)	1.37 (0.41, 4.64)

^1^ Logistic regression modeling odds of irAE outcome, adjusting for age, sex, and ICI therapy; ^2^ additionally adjusted for ECOG score (≥1 vs. 0) and cancer stage (4 vs. <4), sample size n = 103; significant results at alpha = 0.05 are shown in **bold**.

## Data Availability

The data presented in this study are available upon request from the corresponding author due to privacy restrictions.

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
