# Peer review of "Neutrophil to Lymphocyte Ratio as a Biomarker for the Prediction of Cancer Outcomes and Immune-Related Adverse Events in a CTLA-4-Treated Population"

_cancers, 2025, doi:10.3390/cancers17122011_

Round 1
Reviewer 1 Report
Comments and Suggestions for Authors
The study investigates if NLRs can predict immune-related adverse events (irAEs) and treatment response in immunotherapy patients. Patients with multiple irAEs had improved cancer outcomes. Low baseline NLR was linked to higher irAE incidence suggesting NLR could be used as a tool for guiding treatment decisions.
This manuscript would benefit from adding figures or illustrations. Additionally, a flowchart showing the study and results would improve its citability.
On page 7, “Supplementary Materials" - no titles or descriptions.
The paper notes no standard value for low NLR. Values less than 4/5 were chosen based on previous studies. Clarifying why these cutoffs were selected, would be helpful.
The conclusion implies NLR might be a predictive biomarker across ICI classes, though the study focuses only on CTLA-4 inhibition by ipilimumab, which may be an overstatement.
Line 171 (p=?):
Sex differences in dermatologic irAEs show stronger associations with NLR <5 in women. Depicting this data as a table/chart could improve clarity.
The introduction mentions mechanistic pathways. The manuscript would benefit from discussing how CTLA-4 blockade affects T-cell priming versus PD-1’s effect on effector T-cells. Further any insights regarding NLR and mechanisms of its correlation might be helpful
Subgroup analyses have very small groups, leading to wide confidence intervals and reduced statistical power. The paper acknowledges this for rheumatologic irAEs but not for other subgroups. The authors may consider discussing this.
Excluding rare but serious irAEs may bias results toward less severe outcomes, limiting relevance for predicting all irAEs. Relying solely on baseline NLR misses potential dynamic changes that could better predict irAEs or outcomes.
Terminology used in the, eg, "PD1/PD-L1" vs. "PD-1/PDL1" and "rheum/MSK" vs. "rheumatologic and/or musculoskeletal." Please check all for consistency once again
The abstract and simple summary repeat points about NLR predicting irAEs
Author Response
1) The study investigates if NLRs can predict immune-related adverse events (irAEs) and treatment response in
immunotherapy patients. Patients with multiple irAEs had improved cancer outcomes. Low baseline NLR was
linked to higher irAE incidence suggesting NLR could be used as a tool for guiding treatment decisions.
2) This manuscript would benefit from adding figures or illustrations. Additionally, a flowchart showing the study
and results would improve its citability.
a. Thank you for this suggestion. We have added an additional figure to the manuscript (Figure 2) and
Tables. A study flowchart was felt to be challenging as this is a retrospective study and would not add
additional meaning to the manuscript. Additional information regarding study outline and baseline
characteristics of patient cohort was added to sections, Materials and Methods and Results (Tables 1 &
2), respectively.
3) On page 7, “Supplementary Materials" - no titles or descriptions.
a. We appreciate identifying this error, it does not appear the supplemental material was added with the
original submission which was intended. We have organized supplemental material into four tables for
the resubmission upload. Please let us know if supplemental material is not uploaded with
resubmission.
4) The paper notes no standard value for low NLR. Values less than 4/5 were chosen based on previous studies.
Clarifying why these cutoffs were selected, would be helpful.
a. Thank you for the feedback. It is noted in the third paragraph of Materials & Methods that there is
limited literature on establishing NLR cutoffs in patients treated with anti-CTLA-4 immunotherapy
(lines 108-112). Current literature has examined NLR cutoffs in cancer patients treated with anti-
PD1/PD-L1 immunotherapy (references 16-19) and NLR cutoffs of 4 and 5 were what was primarily
used in these studies. In addition, we have added a receiver operating characteristic (ROC) curve
analysis (Figure 2) to assess ideal NLR cutoff to best predict irAEs and we found that NLR 5.09 was the
optimal threshold value for our study cohort (lines 239-244). We have expanded this in discussion
(lines 290-291).
5) The conclusion implies NLR might be a predictive biomarker across ICI classes, though the study focuses only on
CTLA-4 inhibition by ipilimumab, which may be an overstatement.
a. Thank you for this comment. We have edited the conclusion to provide better clarity. Our work is felt
to be hypothesis generating and the intent of the statement was to show that this topic is worth
further study.
6) Line 171 (p=?):
a. Thank you for this comment, this was an error that has been removed.
7) Sex differences in dermatologic irAEs show stronger associations with NLR <5 in women. Depicting this data as a
table/chart could improve clarity.
a. Thank you for the suggestion. As the sex differences among dermatologic irAEs was of very small
numbers and only suggestive (did not reach statistical significance), we have included this in the
supplemental table 3.
8) The introduction mentions mechanistic pathways. The manuscript would benefit from discussing how CTLA-4
blockade affects T-cell priming versus PD-1’s effect on effector T-cells. Further any insights regarding NLR and
mechanisms of its correlation might be helpful
a. We felt this was a great suggestion and have added this information to our introduction (lines 75-82).
9) Subgroup analyses have very small groups, leading to wide confidence intervals and reduced statistical power.
The paper acknowledges this for rheumatologic irAEs but not for other subgroups. The authors may consider
discussing this.
a. Thank you for the feedback. We have changed some of the wording to highlight this limitation in other
irAEs as well in both sections of Results (lines 184-191) and Discussion (lines 265-267) respectively.
10) Excluding rare but serious irAEs may bias results toward less severe outcomes, limiting relevance for predicting all
irAEs. Relying solely on baseline NLR misses potential dynamic changes that could better predict irAEs or
outcomes.
a. Thank you for the comment. We investigated dynamic changes in NLR immediately preceding any irAE
but did not include these data in the manuscript as we did not have enough data for sufficient power.
In the literature, it was common to look at one timepoint, so we proceeded in creating the study
around this one timepoint [11]. Regarding the rare irAEs, we included the rare irAEs in the irAE
definition where they are counted (linear or categorical) (table 1). It was only when evaluating the
organ specific manifestations that we did not include any that were rare with five or fewer as
explained in section Materials and Methods, lines 98-100.
[11] doi:10.1007/s00262-020-02585-w
11) Terminology used in the, eg, "PD1/PD-L1" vs. "PD-1/PDL1" and "rheum/MSK" vs. "rheumatologic and/or
musculoskeletal." Please check all for consistency once again
a. Thank you for the helpful suggestion. We have reviewed again to ensure consistency and edited
accordingly.
12) The abstract and simple summary repeat points about NLR predicting irAEs
a. Thank you for pointing out this duplication. We were hoping the abstract and simple summary could
stand alone in summarizing the manuscript.
Reviewer 2 Report
Comments and Suggestions for Authors
The authors did a review study on the use of neutrophil to lymphocyte ratio to evaluate immune related adverse events triggered by immune checkpoint inhibitor therapy in treatment of melanoma.
Immune checkpoint inhibitors (ICIs) have emerged as a treatment modality for various cancers by enhancing the body's immune response against tumors. The neutrophil-to-lymphocyte ratio (NLR) is recognized as an accessible and cost-effective biomarker that reflects the balance between the innate and adaptive immune responses. In the context of cancer, a low NLR indicates a improved antitumor immunity.
The authors performed a retrospective study evaluating tumor progression and adverse events in patients with melanoma treated with ipilimumab in relation to NLR. Their conclusion underline the importance of NLR to foresee therapy outcome and adverse event: low NLR was associated to many adverse event and best outcomes.
Although the results are convincing and in line with literature data, a couple of corrections seem necessary to me:
1) The study population should include only patients with melanoma treated with ipilimumab monotherapy, excluding those on polytherapy with other ICIs.
2) References are too few: authors should increase citations to best support their work.
Author Response
1) The authors did a review study on the use of neutrophil to lymphocyte ratio to evaluate immune related
adverse events triggered by immune checkpoint inhibitor therapy in treatment of melanoma.
2) Immune checkpoint inhibitors (ICIs) have emerged as a treatment modality for various cancers by enhancing the
body's immune response against tumors. The neutrophil-to-lymphocyte ratio (NLR) is recognized as an
accessible and cost-effective biomarker that reflects the balance between the innate and adaptive immune
responses. In the context of cancer, a low NLR indicates a improved antitumor immunity.
3) The authors performed a retrospective study evaluating tumor progression and adverse events in patients with
melanoma treated with ipilimumab in relation to NLR. Their conclusion underline the importance of NLR to
foresee therapy outcome and adverse event: low NLR was associated to many adverse event and best
outcomes.
4) Although the results are convincing and in line with literature data, a couple of corrections seem necessary to
me:
i. The study population should include only patients with melanoma treated with ipilimumab monotherapy,
excluding those on polytherapy with other ICIs.
a. Thank you for the suggestion. We agree that utilizing data from patients treated with ipilimumab
monotherapy would have reduced potential confounding. However, it was not standard of care at
our site to do monotherapy at the time of the study and therefore we only had a limited number
of patients who were treated with ipilimumab monotherapy. Given this low number of
ipilimumab monotherapy patients, the analysis would not have been powered to make any
conclusion from the data, so we included combination therapy as well.
ii. References are too few: authors should increase citations to best support their work.
a. Thank you for the suggestion, we have reviewed the literature and increased the number of
references.
Reviewer 3 Report
Comments and Suggestions for Authors
- The study makes a valuable contribution by focusing on CTLA-4 inhibitors, which are less frequently studied than PD-1/PD-L1 agents. However, a clear rationale for focusing on ipilimumab specifically, rather than the broader CTLA-4 class or combination therapies, would improve contextual framing.
- The use of both NLR <4 and <5 as cut-offs is well-justified via literature reference, but the decision to use both cut-offs in parallel could benefit from a ROC curve analysis to identify the optimal predictive threshold for this specific cohort.
- The definition and adjudication of irAEs were based on retrospective chart review. While an adjudication panel was used, no grading scale (e.g., CTCAE v5.0) for irAE severity was applied. Including this would allow a more nuanced evaluation of the relationship between irAE severity and outcome.
- The finding that endocrine irAEs were significantly associated with improved outcomes is noteworthy. However, no mechanistic hypotheses or prior literature are referenced to explain why endocrine irAEs specifically might have a stronger prognostic signal than dermatologic or GI irAEs.
- The study includes exploratory interaction testing for sex differences in irAE occurrence, but the statistical significance threshold used (p < 0.10) for interaction effects should be clearly justified, especially since interaction effects are typically underpowered and require cautious interpretation.
- While the association between number of irAEs and improved outcomes is reported, it would be beneficial to present time-to-event analyses (e.g., Kaplan–Meier or Cox models) for progression-free or overall survival, especially given the binary and ordinal classification of outcomes.
- The logistic regression models adjust for age, sex, and ICI therapy, but there is no adjustment for cancer stage, performance status (e.g., ECOG), or baseline LDH—factors known to influence both NLR and immunotherapy outcomes. This may introduce residual confounding.
- The study mentions suggestive associations without statistical significance (e.g., rheum/MSK irAEs and low NLR), but wide confidence intervals indicate limited power in subgroup analyses. Authors should more explicitly state that these results are hypothesis-generating and not confirmatory.
- The data collection period (2004–2017) spans over a decade during which ICI therapy evolved significantly. The authors should comment on potential temporal bias, such as improvements in supportive care or irAE recognition over time that may affect comparability.
- The conclusion suggests NLR as a low-cost predictive biomarker, but the manuscript does not evaluate inter-reader variability in NLR extraction from labs, nor does it address intra-patient variability in NLR levels. Highlighting these limitations would enhance clinical applicability.
Author Response
1) The study makes a valuable contribution by focusing on CTLA-4 inhibitors, which are less frequently studied
than PD-1/PD-L1 agents. However, a clear rationale for focusing on ipilimumab specifically, rather than the
broader CTLA-4 class or combination therapies, would improve contextual framing.
a. Thank you for the comment. Ipilimumab was the only CTLA-4 agent used at our site during the study
time, so we had no other data for other medication of this class. We have added a sentence explaining
this for better clarity (lines 90-91). We also included in the discussion that this was a limitation of the
study and an area for further research (lines 301-303).
2) The use of both NLR <4 and <5 as cut-offs is well-justified via literature reference, but the decision to use both
cut-offs in parallel could benefit from a ROC curve analysis to identify the optimal predictive threshold for this
specific cohort.
a. Thank you for the helpful suggestion. We have added a ROC curve analysis to the manuscript (Figure
2) and we have included it below as well. In addition, we demonstrate that the optimal threshold is in
accordance to literature references, particularly NLR<5.
3) The definition and adjudication of irAEs were based on retrospective chart review. While an adjudication panel
was used, no grading scale (e.g., CTCAE v5.0) for irAE severity was applied. Including this would allow a more
nuanced evaluation of the relationship between irAE severity and outcome.
a. Thank you the comment. We agree that a more nuanced evaluation between irAE severity and
outcome would be of great value to the oncology community. Unfortunately, a formal grading was
not available for a large portion of the irAEs by retrospective chart review. We had concerns about
introducing potential bias should we grade the severity in retrospect. For future research, we would
recommend a prospective study with an independent reviewer to grade such irAEs. We have added an
additional sentence suggesting this as future directions in the discussion (lines 308-310).
4) The finding that endocrine irAEs were significantly associated with improved outcomes is noteworthy.
However, no mechanistic hypotheses or prior literature are referenced to explain why endocrine irAEs
specifically might have a stronger prognostic signal than dermatologic or GI irAEs.
a. Thank you for the comment. There is data suggesting that endocrine irAEs are associated with
improved outcomes (references 29-30, 32). Endocrine irAE mechanism is different and we have
included a discussion on this in both our introduction (lines 80-82) and our discussion (lines 260-265).
5) The study includes exploratory interaction testing for sex differences in irAE occurrence, but the statistical
significance threshold used (p < 0.10) for interaction effects should be clearly justified, especially since
interaction effects are typically underpowered and require cautious interpretation.
a. We agree on the cautious interpretation of effect modifiers through the use of interaction terms. The
threshold p<0.1 is adopted due to the susceptibility of limited power for testing of interaction terms
and is often used as a modestly influential indicator; we have edited the methods to include this
language.
6) While the association between number of irAEs and improved outcomes is reported, it would be beneficial to
present time-to-event analyses (e.g., Kaplan–Meier or Cox models) for progression-free or overall survival,
especially given the binary and ordinal classification of outcomes.
a. Thank you for the suggestion. We initially did not have the dates for mortality; however, have reanalyzed
the patient charts and have now included this in our analysis as shown in Table 4. This table
addresses time-to-event analyses for both time to PD and time to mortality.
7) The logistic regression models adjust for age, sex, and ICI therapy, but there is no adjustment for cancer stage,
performance status (e.g., ECOG), or baseline LDH—factors known to influence both NLR and immunotherapy
outcomes. This may introduce residual confounding.
a. Thank you for this suggestion as well. We also re-analyzed the patient charts to obtain performance
status and cancer stage prior to ICI treatment to include adjustments for this. Table 3, 4, and 5 now
include the updated adjustments for ECOG and cancer staging at time of ICI initiation.
8) The study mentions suggestive associations without statistical significance (e.g., rheum/MSK irAEs and low
NLR), but wide confidence intervals indicate limited power in subgroup analyses. Authors should more explicitly
state that these results are hypothesis-generating and not confirmatory.
a. Thank you for the suggestion, we have added additional lines to make clear that the data are
hypothesis-generating and further studies are needed in the discussion (lines 278-280) and in the
conclusion (lines 317-319).
9) The data collection period (2004–2017) spans over a decade during which ICI therapy evolved significantly. The
authors should comment on potential temporal bias, such as improvements in supportive care or irAE
recognition over time that may affect comparability.
a. Thank you for the suggestion, we have included this potential bias in our discussion section (lines 303-
305).
10) The conclusion suggests NLR as a low-cost predictive biomarker, but the manuscript does not evaluate interreader
variability in NLR extraction from labs, nor does it address intra-patient variability in NLR levels.
Highlighting these limitations would enhance clinical applicability.
a. Thank you for the comment, we have added additional language highlighting the limitation of
potential intra-patient variability (line 108-110). There was some potential for bias as not every
patient had their baseline labs at the exact same time pre-treatment, however on review the vast
majority had the CBC drawn on either the day of treatment or the day prior as noted in Materials and
Methods.
Reviewer 4 Report
Comments and Suggestions for Authors
The study explores the potential of the neutrophil-to-lymphocyte ratio (NLR) as a biomarker to predict immune-related adverse events (irAEs) and clinical outcomes in patients receiving anti-CTLA-4 immunotherapy. Results show that low NLR is linked to increased irAE incidence, multiple irAEs correlate with improved cancer outcomes, and endocrine irAEs are linked to better cancer response. While the topic is relevant—identifying accessible biomarkers for predicting treatment response and toxicity in immunotherapy—the study has several methodological and interpretative limitations that undermine its validity and impact. Below are the major concerns:
- Do not use any abbreviation in the title and the keywords (CTLA-4).
- Page 1, lines 14-42, The simple summary and abstract need substantial modification and improvements. Please maintain the transitions between the simple summary and abstract and used the abbreviated words once when it appears. Moreover, the authors are urged to revise their abstract to improve its originality, practical significance, and prospective uses, while including more precise conclusions and quantitative data.
- Page 1, The authors claim: "Given growing indications for ICI, additional studies are needed to validate such a biomarker in a broader cancer cohort." This statement contradicts the very premise of the study, which itself is limited to a narrow, single-center, melanoma-dominant cohort. If the authors themselves recognize the limitations of generalizing their results, then what exactly is the purpose of submitting this paper?
- The authors have failed to properly contextualize their findings within the broader literature on NLR and irAEs, despite this being an extensively studied area. Numerous recent publications have already investigated and validated the predictive role of NLR across multiple tumor types and ICI regimens, including CTLA-4 inhibitors. The authors are required to explain their novelties in a clear way.
- Page 6, Why were NLR cutoffs (<4, <5) selected arbitrarily instead of using ROC optimization?
- How were missing CBC data handled? Were patients excluded, or was imputation applied?
- Why were rare irAEs (e.g., cardiac, neurological) excluded entirely? Pooling them could have provided preliminary insights.
- Why was no correction for multiple testing applied (e.g., Bonferroni) given the exploratory sex-based analysis?
- How was the adjudication panel’s inter-rater reliability assessed? Without Fleiss’ kappa or similar metrics, subjectivity risks bias.
- In table 1, Why was the cohort split into all patients and a melanoma-only subgroup, and what was the justification for excluding the 11 non-melanoma patients from some analyses?
- Page 7, The conclusion need enhancement. The authors are required to add more findings of their study as well as the main limitations of their work providing future recommendations.
- Finally, the clinical utility of NLR as a predictive biomarker is neither demonstrated nor discussed in a meaningful way. The authors propose that NLR could serve as a low-cost, accessible tool but fail to address critical implementation questions: How would clinicians use this information in practice?
- Double-check for the grammar. Moreover, the authors need to enhance the figure quality which is low-resolution and not readable.
- I highly recommend including a List of Acronyms section to define all the abbreviations used in the work in alphabetical order.
Author Response
1. The study explores the potential of the neutrophil-to-lymphocyte ratio (NLR) as a biomarker to predict immunerelated
adverse events (irAEs) and clinical outcomes in patients receiving anti-CTLA-4 immunotherapy.
2. Results show that low NLR is linked to increased irAE incidence, multiple irAEs correlate with improved cancer
outcomes, and endocrine irAEs are linked to better cancer response.
3. While the topic is relevant—identifying accessible biomarkers for predicting treatment response and toxicity in
immunotherapy—the study has several methodological and interpretative limitations that undermine its validity
and impact. Below are the major concerns:
4. Do not use any abbreviation in the title and the keywords (CTLA-4).
a. Response To editor: Thank you for this suggestion, CTLA-4 is a well-established abbreviation in this
field and has been used in other publications titles (see doi list below)
doi: 10.1182/blood-2017-06-741033
doi: 10.1016/j.intimp.2020.106221
doi: 10.1038/s41585-023-00801-7
We have used the full term at its initial use (the introduction) but have not used this full term for the
title or abstract due to character limits.
5. Page 1, lines 14-42, The simple summary and abstract need substantial modification and improvements. Please
maintain the transitions between the simple summary and abstract and used the abbreviated words once when
it appears. Moreover, the authors are urged to revise their abstract to improve its originality, practical
significance, and prospective uses, while including more precise conclusions and quantitative data.
a. Thank you for this comment. As noted in other reviewer comments, please see our revisions. For the
simple summary we have written this to be stand alone for anyone with or without scientific
background to understand the purpose and finding of the study. Regarding the abstract, we have
adjusted the wording to include more clear language regarding its originality, practical significance,
and prospective uses.
6. Page 1, The authors claim: "Given growing indications for ICI, additional studies are needed to validate such a
biomarker in a broader cancer cohort." This statement contradicts the very premise of the study, which itself is
limited to a narrow, single-center, melanoma-dominant cohort. If the authors themselves recognize the
limitations of generalizing their results, then what exactly is the purpose of submitting this paper?
a. We have removed this sentence in the reformatting of the abstract. Our study is adding to the
literature of potential use of biomarkers. We recognize the limitations of the study but feel our results
are beneficial to the field of investigation of potential biomarkers for identifying outcomes for
patients treated with ICI therapy.
7. The authors have failed to properly contextualize their findings within the broader literature on NLR and irAEs,
despite this being an extensively studied area. Numerous recent publications have already investigated and
validated the predictive role of NLR across multiple tumor types and ICI regimens, including CTLA-4 inhibitors.
The authors are required to explain their novelties in a clear way.
a. Most of the studies have looked at NLR into the context of cancer outcomes but less about prediction
of irAEs. Also, on our review, more of the literature is focused on PD-1/PD-L1 inhibitors and not CTLA-
4 inhibitor, which is the focus of our study.
8. Page 6, Why were NLR cutoffs (<4, <5) selected arbitrarily instead of using ROC optimization?
a. Thank you for the comment. Neither was selected arbitrarily but based on previously published
results as cited in the manuscript (line 136-139). We have added the ROC curve optimization that
identified a value of 5 (5.09) as the optimal threshold (see figure 2) and indicates that the literature
cutoffs are appropriate in this sample.
9. How were missing CBC data handled? Were patients excluded, or was imputation applied?
a. Thank you for the comment. We addressed this in methodology about how CBC data was extracted
and recorded for analysis (line 134-136). There were no patients who had missing CBC data and the
vast majority of CBCs were obtained on either the first day of treatment (just before infusion) or the
day prior.
10. Why were rare irAEs (e.g., cardiac, neurological) excluded entirely? Pooling them could have provided
preliminary insights.
a. Thank you for the comment. We did pool the rare irAEs for the group analysis (where we count total
number or irAEs) but they were excluded from organ specific analyses due to the very low event
occurrence and therefore lack of power. This was addressed in Materials and Methods (lines 98-100).
11. Why was no correction for multiple testing applied (e.g., Bonferroni) given the exploratory sex-based analysis?
a. Thank you for the comment. By definition of an exploratory analysis, multiple corrections are not
usually meaningful and we therefore did not add these corrections.
Turkiewicz A, Luta G, Hughes HV, Ranstam J. Statistical mistakes and how to avoid them -
lessons learned from the reproducibility crisis. Osteoarthritis Cartilage. 2018 Nov;26(11):1409-
1411. doi: 10.1016/j.joca.2018.07.017. Epub 2018 Aug 8. PMID: 30096356.
12. How was the adjudication panel’s inter-rater reliability assessed? Without Fleiss’ kappa or similar metrics,
subjectivity risks bias.
a. Thank you for the suggestion, we have corrected the language in the publication to better reflect the
adjudication process. The panel was an additional physician reviewer (lines 112-113).
13. In table 1, Why was the cohort split into all patients and a melanoma-only subgroup, and what was the
justification for excluding the 11 non-melanoma patients from some analyses?
a. We included the non-melanoma patients for larger group analyses but only looked at melanoma for
the subgroup analyses (e.g. organ specific irAEs) due to the larger sample size for this cancer specific
cohort treated with CTLA-4 inhibitor.
14. Page 7, The conclusion need enhancement. The authors are required to add more findings of their study as well
as the main limitations of their work providing future recommendations.
a. Thank you for the suggestion. As per other reviewers, we have added additional findings in the result
section with Table 4, which is a time-to-event analysis and Figure 2, which analysis NLR for optimal
cutoff to predict irAEs. We have also updated Table 3 and Table 5 to include additional adjustments
for both ECOG and cancer stage. We have updated the discussion to reflect these findings and in
addition further addressed potential limitations and enhanced future directions.
15. Finally, the clinical utility of NLR as a predictive biomarker is neither demonstrated nor discussed in a meaningful
way. The authors propose that NLR could serve as a low-cost, accessible tool but fail to address critical
implementation questions: How would clinicians use this information in practice?
a. Thank you for this suggestion. We have modified the manuscript to highlight the potential benefits of
NLR in the conclusion (line 633-638)
16. Double-check for the grammar. Moreover, the authors need to enhance the figure quality which is lowresolution
and not readable.
a. Thank you for the suggestion, Figure 1 has been updated to enhance resolution and increase
readability of figure legends.
17. I highly recommend including a List of Acronyms section to define all the abbreviations used in the work in
alphabetical order.
a. Thank you for the suggestion, this has been added after the conclusion.
Round 2
Reviewer 2 Report
Comments and Suggestions for Authors
I thank the authors for clarifying my comments. With the addition of new references and the reasons for choosing patients on polytherapy, the work seems complete and worthy of publication.
Reviewer 4 Report
Comments and Suggestions for Authors
The authors have implemented my comments. However, I have minor comments:
1. In the key words you must use a full word of CTLA-4 as follows: cytotoxic T-lymphocyte-associated protein 4 (CTLA-4) blockade or do not mention it. Moreover, in the title still the same question; however, I am leaving it to the editor.
2. I still have this concern about the novelty explanations. Numerous recent publications have already investigated and validated the predictive role of NLR across multiple tumor types and ICI regimens, including CTLA-4 inhibitors. The authors are required to review more related articles showing their main limitations then explain how this work overcomes these gaps. Explain their novelties in a clear way.
3. I highly recommend the authors add a flowchart at the end of the introduction to show the step-by-step process of their work.
4. Double-check the grammar throughout the manuscript.
5. Kindly address these comments diligently.
Thank you for your revision.